# Apamin Enhances Neurite Outgrowth and Regeneration after Laceration Injury in Cortical Neurons

**DOI:** 10.3390/toxins13090603

**Published:** 2021-08-28

**Authors:** Hyunseong Kim, Jin Young Hong, Junseon Lee, Wan-Jin Jeon, In-Hyuk Ha

**Affiliations:** Jaseng Spine and Joint Research Institute, Jaseng Medical Foundation, Seoul 135-896, Korea; biology4005@gmail.com (H.K.); vrt3757@gmail.com (J.Y.H.); excikind@gmail.com (J.L.); poghkl@gmail.com (W.-J.J.)

**Keywords:** bee venom, apamin, cortical neuron, laceration injury, axon regeneration, brain-derived neurotrophic factor, nerve growth factor, regeneration-associated genes

## Abstract

Apamin is a minor component of bee venom and is a polypeptide with 18 amino acid residues. Although apamin is considered a neurotoxic compound that blocks the potassium channel, its neuroprotective effects on neurons have been recently reported. However, there is little information about the underlying mechanism and very little is known regarding the toxicological characterization of other compounds in bee venom. Here, cultured mature cortical neurons were treated with bee venom components, including apamin, phospholipase A2, and the main component, melittin. Melittin and phospholipase A2 from bee venom caused a neurotoxic effect in dose-dependent manner, but apamin did not induce neurotoxicity in mature cortical neurons in doses of up to 10 µg/mL. Next, 1 and 10 µg/mL of apamin were applied to cultivate mature cortical neurons. Apamin accelerated neurite outgrowth and axon regeneration after laceration injury. Furthermore, apamin induced the upregulation of brain-derived neurotrophic factor and neurotrophin nerve growth factor, as well as regeneration-associated gene expression in mature cortical neurons. Due to its neurotherapeutic effects, apamin may be a promising candidate for the treatment of a wide range of neurological diseases.

## 1. Introduction

Apamin is an 18-amino acid peptide and accounts for approximately 2–3% of the dry weight of bee venom [1,2]. Although apamin has long been known as a neurotoxin extracted from bee venom [3,4], research over the past 10 years has uncovered many of its emerging roles and functions, which suggests that apamin is a promising neurotherapeutic agent and acts through modulation of the neuroinflammatory response [5,6,7]. Cumulative evidence on apamin indicates that there is an optimal therapeutic dose range for treatment, above which apamin induces neurotoxicity [8]. Therefore, to avoid cell death, the optimal dose of apamin should be cell-type dependent. Several studies on the use of apamin for the treatment of Alzheimer’s disease (AD) have shown that apamin may enhance neuronal excitability, synaptic plasticity, and organ potentiation in the hippocampal region by blocking the Ca^2+^-activated K^+^ (SK) channels [9,10,11]. Additionally, recent data claim that apamin has a protective effect on dopamine neurons; thus, it can be used to treat degenerative brain disease such as Parkinson’s diseases (PD), as well as regulate synaptic plasticity, memory, and learning [5]. Particularly, apamin has already been patented as a drug used in the treatment of PD [1]. Therefore, apamin may be used as a protective and therapeutic agent against central nervous system (CNS) diseases. However, its effect on mature cortical neurons has not been evaluated. Loss of neurons in the cerebral cortex is a major cause of neurological deficits in adult humans. Thus, a primary cortical neuron has been widely used as a useful in vitro model for neurobiological research. In addition, the advantage of these neurons is that they readily polarize axons and dendrites, and differentiate into mature neurons with axons, dendrites, dendritic spines, and synapses.

In this study, we used primary mature neurons and an in vitro laceration injury model to demonstrate accelerated neurite extension and axon regeneration with apamin treatment in mature neurons. Thus, we precisely demonstrated the neuro-regenerative effects of apamin in enhancing neurite outgrowth and axon regeneration. Together, we analyzed whether apamin alone can induce changes in the expression of neurotrophic factors and regeneration-associated genes (RAGs) to support the broad use of apamin as a therapeutic agent for various neurological diseases.

## 2. Results

### 2.1. Among Bee Venom Components, Apamin Did Not Cause Neurotoxicity in Mature Neurons

Bee venom contains various peptides, and the major components include melittin and phospholipase 2 (PLA2). The schematic presentation summarizes all components of bee venom and its percentage based on the information provided [2] (Figure 1A). We first confirmed the neurotoxicity of bee venom, melittin, PLA2, and apamin in mature cortical neurons using a cell viability assay. We treated day in vitro (DIV) 6 mature cortical neurons with bee venom, melittin, PLA2, and apamin for 24 h. The bee venom and melittin had significant cytotoxic effects in concentrations of ≥0.1 µg/mL (Figure 1B,C) and PLA2 also triggered neuronal death with significantly more neurotoxicity in concentrations from 0.01 µg/mL (Figure 1D). Meanwhile, apamin in concentrations of up to 10 µg/mL was non-toxic to mature cortical neurons (Figure 1E). In a subsequent experiment, we further investigated the cell viability after 48 h of the apamin treatment on DIV6 mature cortical neurons. Apamin in concentrations of up to 10 µg/mL for 48 h was not toxic to mature cortical neurons. In particular, there was a statistically significant increase in apamin concentration from 0.1 to 10 µg/mL (Appendix A, Appendix A). These findings show that apamin is the only safe component of bee venom for primary mature cortical neurons, whereas bee venom, melittin, and PLA2 are toxic to mature cortical neurons.

### 2.2. Apamin May Act as a Potential Contributor to Neurite Outgrowth

We performed a live/dead cell assay to additionally confirm cell viability through the quantification of live (green) and dead (red) cells under the same culture conditions. The cells were mostly observed as green-stained viable cells after apamin treatment at 1 and 10 µg/mL, whereas viable cells were mostly not detected in 1 and 10 µg/mL of bee venom, melittin, and PLA2 (Figure 2A). The fluorescence intensity of the apamin-treated cells stained with green or red dye was quantified and compared to that of the blank group. The green fluorescence intensity was not significantly different between the groups (Figure 2C). In addition, the live/dead assay visually showed that the cells were still viable for 48 h incubation with 1 and 10 µg/mL apamin (Appendix A, Appendix A). Thus, apamin did not exert any cytotoxic effect on neurons, as observed in the CCK assay. We also observed that apamin treatment resulted in longer neurite outgrowth at the single-cell level as well as a dose-dependent increase in the neurite length (Figure 2B). In particular, neurons treated with apamin contained a high concentration of F-actin in the filopodial tips of growth cones (Figure 2D). The results confirmed the safety and efficacy of apamin as a potential contributor to neurite outgrowth.

### 2.3. Apamin Enhances Neurite Outgrowth and Axon Regeneration after Laceration Injury in Mature Cortical Neurons

To examine the effects of apamin on neurite outgrowth and axon regeneration after laceration injury in mature cortical neurons, we observed regenerated axons from the border line at the scratched area using immunocytochemistry. Interestingly, apamin treatment increased axon growth beyond the border line at 24 h post-laceration injury (Figure 3A). A high dose of apamin (10 µg/mL) had a greater effect on axon growth and the total, mean, and maximum neurite length; thus, apamin treatment significantly increased neurite outgrowth in a dose-dependent manner (Figure 3C–E). In addition, we confirmed the long-term effect of apamin on neurite outgrowth and axon regeneration at 48 h after treatment within the scratched area. Longer neurites were observed at 48 h and contained a greater abundance of neuronal axons within the scratched area in the apamin group (Figure 3B). We also examined that apamin promotes F-actin expression in growth cone and induces a significant increase in intensity of F-actin within the scratched area in a dose-dependent manner (Figure 3F). This is a new insight into the neurotherapeutic effects of apamin, showing that it improves neurite outgrowth and axon regeneration after injury.

### 2.4. Apamin Enhances BDNF and NGF Expression after Laceration Injury in Mature Cortical Neurons

We investigated the changes in the expression of brain-derived neurotrophic factor (BDNF) and nerve growth factor (NGF) after apamin treatment in laceration-injured mature neurons. In mature cortical neurons, the relative expression of BDNF decreased significantly after laceration injury compared to the blank group. In contrast, BDNF expression was significantly increased in neurons treated with 1 and 10 µg/mL of apamin compared to the control group (Figure 4A,C). NGF is crucial for neuronal differentiation, maintenance, and regeneration. Here, we found that the expression of NGF showed a tendency to increase with the concentration after apamin treatment but was only significant different between the control and 10 µg/mL of apamin (Figure 4B,D). We further assessed the BDNF and NGF protein expression using western blot analyses (Figure 4E). Both BDNF and NGF protein levels were increased by apamin treatment in a dose-dependent manner, and quantification was performed using flow cytometry. BDNF-positive neurons were significantly increased after apamin treatment in a dose-dependent manner, reaching a maximum of approximately 3% (Appendix A, Appendix A). Furthermore, similar to the results of BDNF, NGF-positive neurons by flow cytometry showed that the cells were significantly increased after apamin treatment in a dose-dependent manner, as shown in Appendix A.

### 2.5. Apamin Promotes the Expression of Growth-Associated Genes in Mature Cortical Neurons

We further analyzed the changes in the expression of growth-associated genes to better understand the mechanisms underlying improved neurite outgrowth and axon regeneration after apamin treatment in mature neurons. The gene expression levels of *BDNF* and *NGF* were increased in the apamin groups compared to the blank group (Figure 5A,B). In particular, *NGF* expression significantly increased in an apamin concentration-dependent manner, and expression of one of the regeneration-associated genes, neurofilament 200-kDa (*NF200*), was significantly increased in the apamin groups; growth-associated protein (*GAP43*) expression was significantly different between the blank and 10 µg/mL apamin groups (Figure 5C,D). 

## 3. Discussion

Although apamin is known to be a neurotoxic polypeptide, recent findings have reported that apamin potentially elicits important neuroprotective roles relevant to the treatment of CNS diseases [5,12]. Interestingly, pharmacological actions of apamin have been attributed to apoptosis, fibrosis, and various diseases [13]. A recent study showed that apamin was not cytotoxic to BV2 microglial cells at 0.5–5 µg/mL and can suppress LPS-induced neuroinflammatory responses by regulating SK channels and TLR4-mediated signaling pathway [7]. Particularly associated with SK channels, apamin has long been considered to specifically block the SK channel as a selective antagonist. SK channels are widely associated with the central nervous system and play important roles in synaptic plasticity, learning, and memory regulation [1]. Specifically, SK channels are present in midbrain dopaminergic neurons and regulate the spontaneous activity of dopamine neurons [14]. Based on these findings, apamin contributes to learning and memory control by selectively blocking SK cannels. Previous studies have revealed that apamin has shown neuroprotective effects on mesencephalic or midbrain dopaminergic neurons in culture [15,16]. However, there is not enough scientific information to determine an optimal therapeutic dose range for the treatment of neurological diseases. Bee venom has traditionally been regarded as an ancient medicine used in the treatment of a variety of diseases, including arthritis, back pain, cancerous tumors, and multiple sclerosis [17,18,19]; apamin is the third most abundant main element in bee venom [2]. For bee venom therapy, it is crucial that the specific concentration used not cause cellular toxicity in patients. Therefore, recent studies have examined the toxicity of bee venom, including experimental descriptions of optimal doses [20]. We report here for the first time that apamin is a promising agent of axonal regeneration-promoting neurotherapy without neurotoxicity in doses of up to 10 µg/mL in mature cortical neurons. Meanwhile, other components of bee venom, such as melittin and PLA2, began to cause neurotoxicity in cortical neurons in a concentration of 0.1 µg/mL. Collectively, we suggest that the optimal therapeutic dose range of apamin be 1–10 µg/mL for neurite extension and outgrowth. Besides, cultured cortical neuron showed more abundant F-actin-expressing growth cone on the ends of growing axons after apamin treatment and laceration injury. Growth cones are F-actin-rich structures at the tip of an extending axon that regulate the extension of a regenerating neurite and act as guidance cues during development [21]. Therefore, apamin enhances F-actin content in the growth cone to stimulate axon growth. Furthermore, the findings described here show that apamin alone in optimal doses can increase gene expressions of *BDNF* and *NGF*, as well as regeneration-associated genes *NF200* and *GAP43.* However, we only confirmed this with apamin treatment after laceration that showed an increase in both the neurite outgrowth and axon regeneration in the lacerated area. This mechanical scratch injury provides a suitable traumatic in vitro model for studying neuronal injuries induced due to spinal cord injury or traumatic brain injury; however, the extent of neuronal injury is limited to the lacerated area. Therefore, in vitro models of global neuronal injury are needed to delineate the underlying pathophysiological mechanisms of how apamin potentially affects neuronal physiology and function. In addition, future studies should aim to examine the therapeutic effect of apamin in neurological animal models and determine whether the increased expression of specific genes and proteins by apamin can induce functional recovery after injury. 

## 4. Materials and Methods

### 4.1. Primary Cultures of Rat Cerebral Cortical Neurons

All animals used in this study were maintained in accordance with the guidelines of the Jaseng Animal Care and Use Committee (JSR-2020-03-004). Primary cortical neurons were obtained from Sprague-Dawley rat embryos (embryonic day 17, Daehan Bio Link, Chungbuk, Korea) as previously described [22]. In brief, a pregnant female rat was sacrificed by CO_2_-induced asphyxiation, followed by immediate separation of the embryos via a cesarean section using large scissors and toothed forceps. The embryos were placed in a 100 mm × 20 mm petri dish containing cold Hank’s balanced salt solution (HBSS; Gibco BRL, Grand Island, NY, USA) on ice. The skin on the head and skull of the embryo were carefully removed using No. 5 fine forceps until the upper surface of the brain was visible. The cerebral cortex was carefully isolated and placed in HBSS, and the meninges were manually removed from the cerebral hemispheres. The tissues were rinsed twice in HBSS, and digested with the Neural Tissue Dissociation kit (Miltenyi Biotec, Bergisch Gladbach, NRW, Germany) and GentleMACStm Octo Dissociator (Miltenyi Biotec) for 20 min at 37 °C; then, the supernatant was discarded. Subsequently, the tissues were rinsed twice in 2 mL HBSS and centrifuged at 2000 rpm for 3 min to obtain the cell pellet. Cells were triturated in 1 mL of cortical neuron culture medium containing neurobasal medium (Gibco BRL) supplemented with 1% penicillin/streptomycin (Gibco BRL), 1% Gluta-MAX (Gibco BRL), and 2% B27 (right before using, Gibco BRL). Single cells were then seeded onto coated 12 mm glass coverslips (Paul Marienfeld GmbH & Co., Lauda-Königshofen, Germany) in 24-well plates at 2 × 10^4^ cells/500 µL for immunocytochemistry, 6-well plates at 2 × 10^6^ cells/2 mL for flow cytometry, 60 mm^2^ dishes at 4 × 10^6^ cells/3 mL for real-time PCR, and 96-well plates at 2 × 10^4^ cells/100 µL for cell viability assay, respectively, which were then coated with 20 mg/mL poly-D-lysine (Gibco BRL) overnight, followed by 10 µg/mL laminin (Sigma-Aldrich, St. Louis, MO, USA) for 2 h at 4 °C. Bee venom (Chungjin Biotech, Ansan, Korea), melittin (Sigma-Aldrich), PLA2 (Sigma-Aldrich), and apamin (Sigma-Aldrich) were treated for 24 or 48 h for cell viability assay. 

### 4.2. Neuronal Viability Assays

Neuronal viability was evaluated by Cell Counting Kit-8 assay (CCK-8; Dojindo, Ku-mamoto, Japan) and with a live/dead cell imaging kit (Thermo Fisher Scientific, Waltham, MA, USA). First, the cells were added to a 96-well plate for the CCK assay and treated with various concentrations of bee venom, melittin, PLA2, and apamin (0.001, 0.01, 0.1, 1, and 10 µg/mL). After incubation for 24 or 48 h, CCK-8 solution was added to each well in a ratio of 10/~1 of the culture medium. After 4 h, absorbance was measured at 450 nm using a microplate reader (Epoch, BioTek, Winooski, VT, USA). Cell viability was calculated as the percentage of surviving neuron cells relative to the value of the blank group. A live/dead cell imaging kit (Invitrogen, Grand Island, NY, USA) was used to visualize and verify neural viability. The dyeing solution contains two probes: calcein AM, which marks living cells as green, and ethidium homodimer-1, which marks dead cells as red. The cells were prepared on each coverslip in a 24-well plate and treated with 1 or 10 µg/mL of bee venom, melittin, PLA2, and apamin. After the cells were incubated for 24 or 48 h, the culture medium was replaced with cortical neuron medium containing a dye solution and incubated at 37 °C for 15 min. After dyeing, the samples were rinsed in PBS and mounted with fluorescence mounting medium (Dako Cytomation, Carpinteria, CA, USA). All images were randomly captured at 100× and 400× magnification with a confocal microscope (Eclipse C2 Plus, Nikon, Minato City, Tokyo, Japan). Live/dead cells were quantified by measuring the intensities of green- and red-stained cells using ImageJ software (1.37 v, National Institutes of Health, Bethesda, MD, USA).

### 4.3. Laceration Injury 

Laceration injury has been previously described in detail [23]. Cortical neurons were cultured in a neuronal culture medium on poly-D-lysine- and laminin-coated 12 mm coverslips inside 24-well culture plates until DIV6. Neurites were then mechanically wound by dragging a 10 µL pipette tip centrally across the coverslip, followed by treatment with apamin at 1 and 10 µg/mL, and fixation of the cells after 24 or 48 h. The experimental design showing the timeline of the procedures is described in Figure 6. 

### 4.4. Immunocytochemistry

We performed immunocytochemistry to quantify the neurite outgrowth and axon regeneration in the cultured cortical neurons. The samples were analyzed after 24 or 48 h of the apamin treatment (1 or 10 µg/mL) in lacerated cortical neurons. In brief, the samples were fixed with 4% paraformaldehyde for 30 min, rinsed three times with PBS for 5 min per rinse, and then incubated with 0.2% triton X-100 in PBS for 5 min. After two rinses with PBS for 5 min and blocking with 2% normal goat serum in PBS for 1 h, the samples were incubated with primary antibodies diluted in 2% normal goat serum overnight at 4 °C. The primary antibodies used were as follows: brain-derived neurotrophic factor (BDNF; 1:200; Abcam, Cambridge, MA, USA), nerve growth factor (NGF; 1:100; Abcam), Tuj1 (1:2000; R&D systems, Minneapolis, MN, USA), and rhodamine phalloidin (F-actin; 1:1000; Invitrogen). After three washes with PBS for 5 min per wash, the samples were incubated with fluorescent conjugated secondary antibodies (FITC-conjugated goat anti-rabbit IgG, FITC-conjugated goat anti-mouse IgG, Rhodamine Red-X-conjugated goat anti-rabbit IgG, Rhodamine Red-X-conjugated goat anti-mouse IgG, Jackson Immuno-Research Labs, West Grove, PA, USA) diluted at 1:300 in 2% normal goat serum for 2 h. After 2 h of incubation at room temperature, the cells were washed three times for 5 min with PBS. The samples were treated with 4-6-dia-midino-2-phenylindole (DAPI; Tokyo Chemical Industry Co., Tokyo, Japan) containing PBS for 10 min at room temperature. Subsequently, the cells were washed three times with PBS for 5 min, mounted with fluorescence mounting medium (Dako Cytomation, Carpinteria, CA, USA), and imaged using a confocal microscope (Eclipse C2 Plus, Nikon). To quantify neurite outgrowth, we first captured the 10 representative images at 400× magnification using confocal microscopy with fixed acquisition settings and then we used three parameters, the total, mean, and maximum neurite outgrowth, using ImageJ software (1.53 v, Fiji Distribution, National Institute of Health, Bethesda, MD, USA). In addition, the average intensity was compared quantitatively through ImageJ software using confocal images captured under the same acquisition settings.

### 4.5. Western Blotting

Total proteins were extracted in each group using RIPA buffer (CellNest, Minato, Tokyo, Japan) with Protease Inhibitor Cocktail Set III (1:1000, Millipore, Billerica, MA, USA). Protein concentration was measured using the Pierce BCA Protein Assay Kit (Thermo Fisher Scientific), in accordance with the manufacturer’s protocol. Protein samples were separated by SDS-PAGE, transferred to a polyvinylidene difluoride (PVDF) membrane, blocked with 5% DifcoTM skim milk (BD Biosciences, Franklin Lakes, NJ, USA) in 1× Tris-buffered saline (TBS, Bio-Rad, Hercules, CA, USA) with 0.1% Tween 20 (Sigma-Aldrich), and probed using various antibodies. The Western blots were visualized using ECL (Bio-Rad) and exposed to the Amersham Imager 600 (GE Healthcare Life Sciences, Uppsala, Sweden). The equivalence of protein loading was verified by probing for actin. The antibodies used were as follows: rabbit anti-BDNF (1:500, Abcam), rabbit anti-NGF (1:200, R&D Systems), mouse anti-β-actin (1:1000, Santa Cruz Biotechnology, Santa Cruz, CA, USA), horseradish peroxidase-conjugated anti-rabbit antibody (1:2500, Abcam), and horseradish peroxidase-conjugated anti-mouse antibody (1:2500, Abcam).

### 4.6. Real-Time PCR

We assessed changes in *BDNF*, *NGF*, *NF200*, and *GAP43* at the mRNA level after apamin treatment via real-time polymerase chain reaction (PCR). In brief, the samples were prepared as described and the total RNA was isolated using Trizol reagent (Thermo Fisher Scientific); cDNA was synthesized using random hexamer primers and Accupower RT pre-mix (Bioneer, Daejeon, Korea). All primer pairs were designed using UCSC Genome Bioinformatics and the NCBI database (Table 1). Real-time PCR was performed using iQSYBR green Supermix (Bio-Rad) on a CFX Connect Real-Time PCR Detection System (Bio-Rad) under the following conditions: an initial cycle of 3 min at 95 °C, followed by 45 cycles of 15 s at 95 °C, and finally 30 s at 60 °C. Each assay was performed at least three times. Target gene expression was normalized to GAPDH levels and expressed as a fold-change relative to the control.

### 4.7. Flow Cytometry 

The flow cytometric assay was used to quantify the BDNF- and NGF-positive cells in each group. Briefly, the cells were isolated with a cell scraper and centrifuged at 2000 rpm for 3 min. The cells were then incubated with anti-BDNF (1:100, Abcam) or anti-NGF (1:50, Abcam) with FITC-Goat anti-Rabbit Ab (1:200, Jackson Immuno-Research Labs) in binding buffer (BD science, Franklin Lakes, NJ, USA) for 10 min and then analyzed directly using FACS (Accuri C6 plus flow cytometer, BD Bioscience) after adding the same volume of PBS.

### 4.8. Statistics

All results are expressed as the mean ± standard error of the mean (SEM). Comparisons among each group were analyzed using one-way analysis of variance (ANOVA) with Tukey’s post-hoc test (Graph-Pad Prism, California, CA, USA). Differences were considered statistically significant based on the following comparisons: * *p* < 0.05, ** *p* < 0.01, *** *p* < 0.001, or **** *p* < 0.0001 vs. the blank or control group.

## Figures and Tables

**Figure 1 toxins-13-00603-f001:**
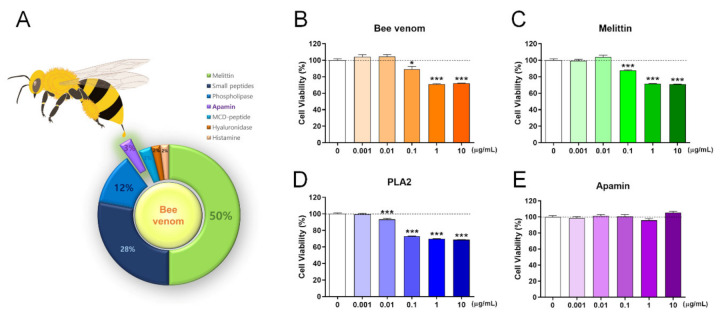
Cell viability assays of bee venom and its components such as melittin, PLA2, and apamin using the cell counting kit (CCK) assay after 24 h from treatment in DIV6 mature cortical neurons. (**A**) Schematic presentation of bee venom composition. (**B**) CCK assay of neuronal cell viability after treatment with (**B**) bee venom, (**C**) melittin, (**D**) PLA2, and (**E**) apamin after 24 h of incubation with cortical neurons. Data are expressed as the mean ± standard error of the mean (SEM). Significant differences indicated as * *p* < 0.05 and *** *p* < 0.001 vs. the blank group, as analyzed by the one-way analysis of variance (ANOVA) and Tukey’s post-hoc test.

**Figure 2 toxins-13-00603-f002:**
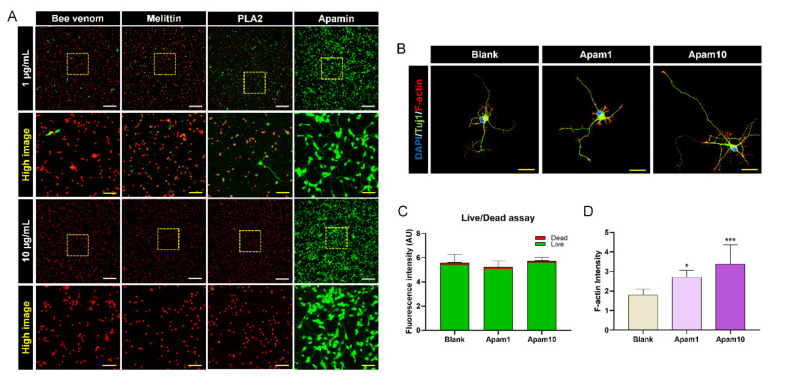
Live/dead assay and the analysis of F-actin expression after 24 h of apamin treatment on DIV6 mature cortical neurons. (**A**) Representative images of viable (green) and nonviable (red) neurons after bee venom, melittin, PLA2, and apamin treatment of DIV6 mature cortical neurons. White scale bar = 200 µm. Yellow scale bar = 50 µm. (**B**) Representative fluorescence images of neuronal morphology and outgrowth using staining for Tuj1 (green), F-actin (phalloidin, red), and 4′6-diamidino-2-phenylindole (DAPI; blue) after apamin treatment in DIV6 mature cortical neurons without laceration injury. Yellow scale bar = 20 µm. (**C**) Graph showing live/dead analysis results for viable (green) and nonviable (red) neuronal cell fluorescence intensities. (**D**) The relative fluorescence intensity of F-actin at the single-cell level after apamin treatment. Data are expressed as the mean ± standard error of the mean (SEM). Significant differences indicated as * *p* < 0.05 and *** *p* < 0.001 vs. the blank group, as analyzed via one-way analysis of variance (ANOVA) with Tukey’s post-hoc test.

**Figure 3 toxins-13-00603-f003:**
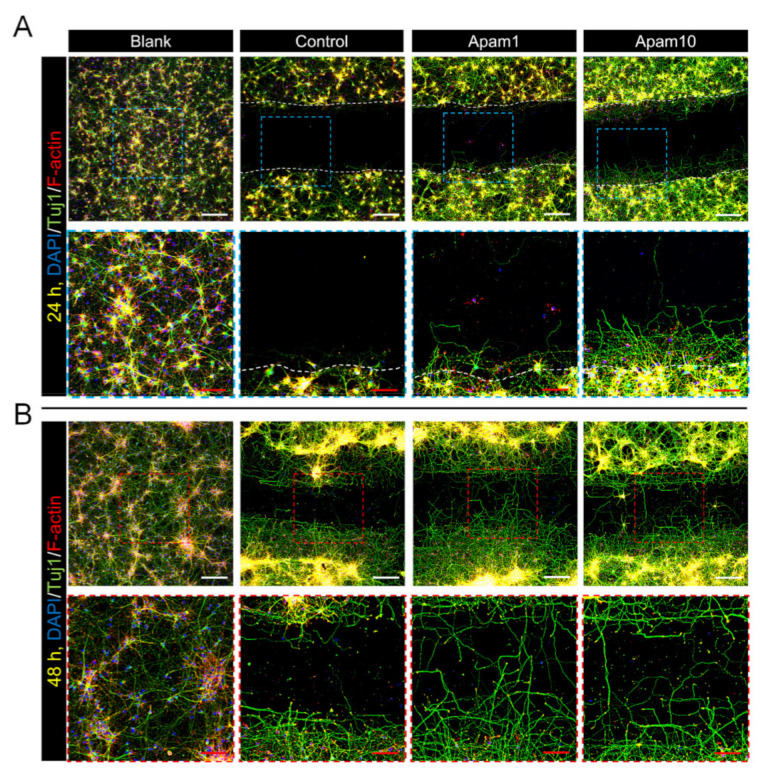
Effect of apamin on neurite outgrowth and axon regeneration at 24 and 48 h after laceration injury in DIV6 mature cortical neurons. (**A**,**B**) Representative fluorescence images of neurite outgrowth and axon regeneration stained for Tuj1 (green), F-actin (phalloidin, red), and DAPI (blue) at 24 h and 48 h after apamin treatment and laceration injury in DIV6 mature cortical neurons. White scale bar = 200 µm. Red scale bar = 50 µm. (**C**–**F**) Quantification results of (**C**) total neurite length, (**D**) mean neurite length, and (**E**) maximal neurite length in each group at 24 h after laceration injury within scratched area and (**F**) F-actin intensity at 48 h after laceration injury within scratched area. Data are expressed as the mean ± standard error of the mean (SEM). Significant differences indicated as * *p* < 0.05, ** *p <* 0.01, *** *p* < 0.001, and **** *p* < 0.0001 vs. the control group, as analyzed via one-way analysis of variance (ANOVA) with Tukey’s post-hoc test.

**Figure 4 toxins-13-00603-f004:**
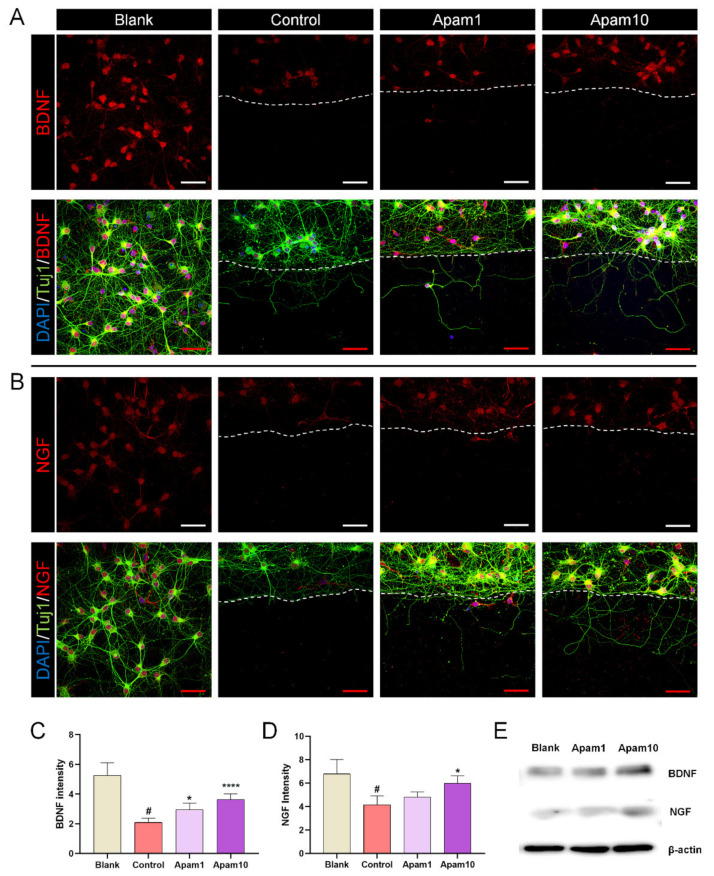
Effect of apamin on BDNF and NGF expression at 24 h after laceration injury in DIV6 mature cortical neurons. (**A**,**B**) Representative fluorescence images of expression of neurotrophic factors, stained for Tuj1 (green), BDNF (red), and DAPI (blue), or Tuj1 (green), NGF (red), and DAPI (blue), and laceration injury in DIV6 mature cortical neurons. White scale bar = 200 µm. Red scale bar = 50 µm. (**C**,**D**) Quantification intensities of (**C**) BDNF and (**D**) NGF in each group. (**E**) Western blot analysis of BDNF and NGF after 1 and 10 µg/mL of apamin treatment in mature cortical neurons. Data are expressed as the mean ± standard error of the mean (SEM). Significant differences indicated as ^#^
*p* < 0.001 compared with the blank group; * *p* < 0.05 and **** *p* < 0.0001 vs. the control group, as analyzed via one-way analysis of variance (ANOVA) with Tukey’s post-hoc test.

**Figure 5 toxins-13-00603-f005:**
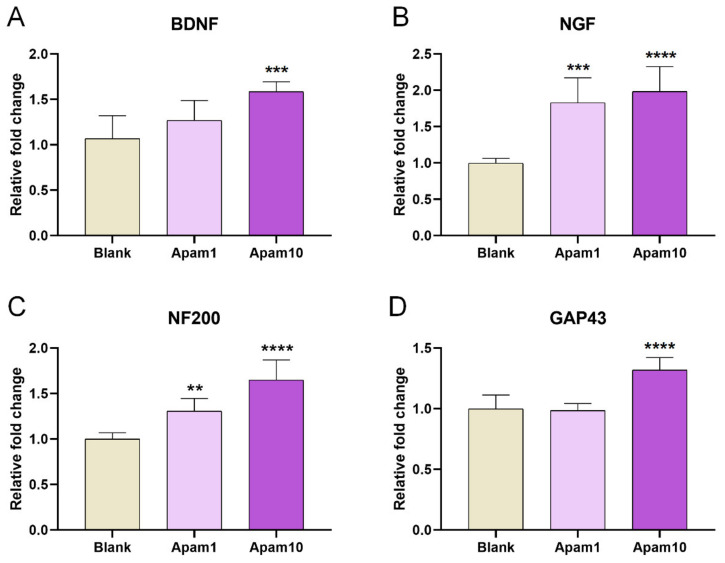
Real-time PCR analysis for (**A**) *BDNF*, (**B**) *NGF*, (**C**) *NF200*, and (**D**) *GAP43* after 24 h of the apamin treatment in DIV6 mature cortical neurons. Data are expressed as the mean ± standard error of the mean (SEM). Significant differences indicated as ** *p* < 0.01, *** *p* < 0.001, and **** *p* < 0.0001 vs. the blank group, as analyzed via one-way analysis of variance (ANOVA) with Tukey’s post-hoc test.

**Figure 6 toxins-13-00603-f006:**
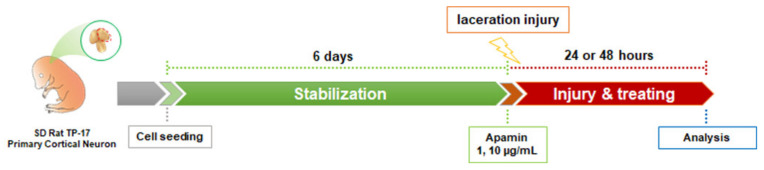
Schematic timeline of the experimental procedures in laceration injury.

**Table 1 toxins-13-00603-t001:** Primer sequences used for real-time polymerase chain reaction (PCR) analysis.

Gene	5′–3′	Primer Sequence
*BDNF*	Forward	CTTGGAGAAGGAAACCGCCT
	Reverse	GTCCACACAAAGCTCTCGGA
*NGF*	Forward	CCAAGGACGCAGCTTTCTATC
	Reverse	CTGTGTCAAGGGAATGCTGAAG
*NF200*	Forward	AACACCACTTAGATGGCGGG
	Reverse	ACGTGGAGCGTTCAGCAATA
*GAP43*	Forward	TGCCCTTTCTCAGATCCACT
	Reverse	TTGCCACACAGAGAGAGAGG
*GAPDH*	Forward	CCCCCAATGTATCCGTTGTG
	Reverse	TAGCCCAGGATGCCCTTTAGT

## Data Availability

The data presented in this study are available on request from the corresponding author.

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
