# Peer review of "Apamin Enhances Neurite Outgrowth and Regeneration after Laceration Injury in Cortical Neurons"

_toxins, 2021, doi:10.3390/toxins13090603_

Round 1

Reviewer 1 Report

In general, the manuscript is well written, the objective of the research is clear, and the methodological design is detailed.

This is a very interesting manuscript that shows the neuro-regenerative effects of apamin in enhancing neurite outgrowth and axon regeneration, using DIV6 mature cortical neurons and an ex vivo laceration injury model. Additionally, it has been shown that apamine could induce the upregulation of BDNF, NGF and regeneration-associated genes expression in cortical neurons.

I believe the manuscript has the suitable scope for the Toxins. I only have some questions/requests.

  1. Abstract (line 10): The sentence “Apamin was the only component that did not cause neurotoxicity through a cell viability assay” is not correct, since only mellitin, PLA2 and apamine were analyzed individually by the cell viability assay but bee venom has many other components. Improve sentence writing
  2. Was the composition of the bee venom shown in Figure 1A determined by the authors in this paper? If the answer is negative, the authors must add the citation of the article from which the data to prepare the figure were obtained.
  3. Caption of Figure 1A: I think "Schematic characterization of ..." should be replaced by "Schematic presentation of ...". Present the meaning of NS at the end of the figure 1 caption.
  4. Figure 2. Live/dead assay and the analysis of F-actin expression after apamin treatment Figure 24. h on DIV6 mature cortical neurons.”. Correct this sentence (after 24 h of the apamin treatment…)
  5. Discussion, line 156: replace "ml' with "mL"
  6. Discussion, lines 157-168: “We report here for the first time that apamin is the only bee venom peptide that did not cause neurotoxicity in mature cortical neurons”. Correct this sentence. Please see the request number 1 above.
  7. I think it would be interesting to discuss whether the neurotherapeutic effect of apamine is related to its interaction with potassium channels.

Author Response

We wish to re-submit the manuscript titled “Apamin Enhances Neurite Outgrowth and Regeneration After Laceration Injury in Cortical Neurons” The manuscript ID is 1327124.

We thank the editor and the reviewers for their excellent and constructive comments, which clearly helped to improve the quality of this manuscript. We have performed additional experiments as detailed below, thereby addressing the issues raised by the reviewers. We are pleased to provide the following point-by-point reply. Appropriate changes, suggested by the reviewers, have been introduced to the manuscript (highlighted within the manuscript). We hope that our manuscript will be acceptable for publication in Toxins.

Reviewer 2 Report

This manuscript reported that apamin at two concentrations could exert neuroprotective effects on cultured neurons through cell assay. It further suggested that such neuroprotection was mediated by BDNF, NGF and other factors. Though the results are potentially interesting, several major weaknesses remain.

Major comments:

  1. The major concern about this study is the mechanistic insight. The authors claimed that BDNF (and NGF et al.) could mediate the apamin-triggered neuroprotection. This was not supported by any experimental evidence presented. Neither pharmacological tools or knock-down methods targeting these factors were used to demonstrate directly in these cells that apamin protect the neurons through these factors. This should be addressed in a convincing way.
  2. In figure 3, apamin induced neurite outgrowth and axon regeneration at 48 h more obviously than 24 hours. This make the experiments in Figure 1 quite questionable, as the authors only tested whether apamin triggered cell viability for 24 hours. If the experimental time were extended 48 hours, the conclusion could be different.

Other comments

  1. In figure 1, the representative pictures for the cell viability should be presented to allow the readers to judge the quality of the experiments.
  2. In figure 1, both melittin and PLA2 at either 1 or 10 microgram per ml induced the cell viability at roughly 60-70%, but the total bee venom also induce a similar amount cell viability instead of a lower cell viability. This discrepancy needs to be explained.
  3. In figure 2A, the individual cell could not be clearly observed. The authors should make some representative cells here bigger so that the readers could judge the cell quality.
  4. Figure 3C is too small to be examined clearly. The authors should make the texts throughout all figures consistently big and clear.

Author Response

(The authors gave the same response as above.)

Round 2

Reviewer 2 Report

The revision is satisfactory. It is suitable for publication now.